# Excitations in a superconducting Coulombic energy gap

Juan Carlos Estrada Saldaña [1✉], Alexandros Vekris[1,2], Luka Pavešić [3,4], Peter Krogstrup[1], Rok Žitko [3,4], Kasper Grove-Rasmussen[1] & Jesper Nygård [1✉]

Cooper pairing and Coulomb repulsion are antagonists, producing distinct energy gaps in superconductors and Mott insulators. When a superconductor exchanges unpaired electrons with a quantum dot, its gap is populated by a pair of electron–hole symmetric Yu-Shiba-Rusinov excitations between doublet and singlet many-body states. The fate of these excitations in the presence of a strong Coulomb repulsion in the superconductor is unknown, but of importance in applications such as topological superconducting qubits and multi-channel impurity models. Here we couple a quantum dot to a superconducting island with a tunable Coulomb repulsion. We show that a strong Coulomb repulsion changes the singlet many-body state into a two-body state. It also breaks the electron–hole energy symmetry of the excitations, which thereby lose their Yu-Shiba-Rusinov character.

[1] Center for Quantum Devices, Niels Bohr Institute, University of Copenhagen, 2100 Copenhagen, Denmark. [2] Sino-Danish College (SDC), University of Chinese Academy of Sciences, Beijing, China. [3] Jožef Stefan Institute, Jamova 39, SI-1000 Ljubljana, Slovenia. [4] Faculty of Mathematics and Physics, University of Ljubljana, Jadranska 19, SI-1000 Ljubljana, Slovenia. ✉email: juan.saldana@nbi.ku.dk; nygard@nbi.ku.dk

In a large superconductor, an adsorbed spin impurity binds to a quasiparticle screening cloud to form a state known as the Yu-Shiba-Rusinov (YSR) singlet, whose excitation energy with respect to the unbound doublet is below the superconducting energy gap, $\Delta$[1]. The miniaturization of the superconductor into an island reduces charge screening and introduces an energy gap for the addition of electrons, the Coulomb repulsion, $E_c$ (see Fig. 1a)[2,3], with yet unexplored consequences on the ground state and the subgap spectrum. Such exploration is of relevance in the study of magnetic impurities adsorbed to superconducting droplets[4,5], in quantum-dot (QD) readout of Majorana qubits based on superconducting islands[6–8], and in realizations of superconducting variants of the multichannel Kondo model[9–11].

In the absence of a spin impurity, the charging of a superconducting island (SI) depends on the ratio $E_c/\Delta$, with $E_c/\Delta < 1$ leading to Cooper pair ($2e$) charging and $E_c/\Delta > 1$ to $1e$ charging[2,3]. In the latter case, even numbers of electrons condense as Cooper pairs, while a possible odd numbered extra electron must exist as an unpaired quasiparticle[3].

**Fig. 1 From Yu-Shiba-Rusinov to Coulomb-influenced excitations.**
**a** Idealized system displaying Coulomb-influenced subgap excitations. An impurity with Coulomb repulsion $U$, carrying a spin degree of freedom when occupied by one electron, is coupled with hybridization $\Gamma$ to a superconducting island with Coulomb repulsion $E_c$ and energy gap $\Delta$. A quasiparticle is plucked away from the Cooper pair condensate to form a YSR singlet bound state with the spin, with the competing doublet state destabilized by $E_c$. In a device, the impurity can be a quantum dot, and the QD and SI gate-induced charges $\nu$ and $n_0$ can be tuned with gate voltages. **b**, **c** Calculated charge parabolas versus $\delta n_0$, which is $n_0$ referenced to an even integer value (with $\nu = 1$). **d**, **e** Same as **b** and **c**, but now sweeping $\nu$ (with $\delta n_0 = 0$). $E_c = 0$ in **b**, **d** and $E_c = 1.3\Delta$ in **c**, **e**. $\Gamma/U = 0.05$ in **b–e**. The parabolas are tagged by their total charge $n_{GS}$ referenced to an even integer value. A green dot indicates the destabilization of the doublet state by $E_c$ for $n_0 = 1$ from **b** to **c**. Red (blue) arrows indicate addition (removal) excitations. For simplicity, continuum parabolas are not included.

Here we provide the first spectral evidence of the many-body excitations in a superconducting Coulombic gap. The spin impurity resides in a gate-defined QD in an InAs nanowire, and the SI is an Al crystal grown on the nanowire with gate-tunable Coulomb repulsion. Both QD-SI and SI-QD-SI devices are investigated in this work. We demonstrate that a strong Coulomb repulsion forces exactly one quasiparticle in the SI to bind with the spin of the QD in the singlet ground state (GS). The Coulomb repulsion also enforces a positive-negative bias asymmetry in the position of the excitation peaks which is uncharacteristic of YSR excitations.

## Results

**Excitations in a quantum dot coupled to a superconducting island.** Figure 1b–e summarize the energy dispersions which can arise when a QD is coupled to a superconductor. In Fig. 1b, the usual YSR case ($E_c = 0$) with the QD gate-induced charge tuned to $\nu = 1$ is depicted. The doublet GS and singlet excited state energies are independent of the gate-induced charge in the superconductor, $n_0$, and excitations between these two states are electron-hole symmetric[12]. As shown in Fig. 1c, introducing $E_c > 0$ in the superconductor produces a parabolic dispersion distorted by the hybridization ($\Gamma$) between the QD and the SI, which couples states of the same total charge. For odd $n_0$, the energy of the doublet state is increased by $\approx E_c$ (green dot), while for even $n_0$ it is the energy of the singlet state which is penalized by this amount. For odd $n_0$ and $E_c > \Delta$, the GS is a singlet even if $\Gamma \to 0$. For $E_c < \Delta$, the singlet can be the GS if the YSR binding energy $E_B$ is large enough so that $E_c > \Delta - E_B$, which is achieved by increasing $\Gamma$[13].

Due to $U > 0$, the dispersion against $\nu$ is approximately parabolic in both the $E_c = 0$ (up to a constant) and $E_c > 0$ cases, as shown in Fig. 1d, e. For $E_c = 0$ (Fig. 1d), the electron and hole excitations are symmetric due to the degeneracy of the even-parity parabolas. This ceases to be the case for $E_c > 0$ (Fig. 1e). The asymmetry is maximal in the absence of additional QD levels. For $\nu > 1$ ($\nu < 1$), an extra electron (hole) must be stored in the SI with excitation energy $\Delta + E_c$, but an extra hole (electron) can be added to either the QD or the SI, leading to a superposition of states with excitation energies $-(\epsilon_d + U)$ and $-(\Delta + E_c)$, where $\epsilon_d$ is the energy level of the QD (details on Supplementary Fig. 1). The extra electron or hole either forms a quasiparticle or a Cooper pair, depending on the parity of the SI occupation of the initial state. Superconducting Coulombic excitations (SCE) are only symmetric at the special gate points where the excited parabolas cross each other.

Our QD-SI device (Fig. 2a, b) is modeled as in the scheme shown in Fig. 2c[13]. The SI is conceived as several hundreds of electronic levels. Its charge is tuned by $n_0$, equivalent to top gate voltage $V_S$ in the device. The corresponding Hamiltonian includes pairing between time-reversed states to produce the superconducting gap, $\Delta$, and coupling to the QD, $\Gamma$, which is tuned by top gate voltage $V_3$ in the device. We consider constant Coulomb interactions $U$ for the QD, $E_c$ for the SI, and $V$ for the interdot charging due to the QD-SI inter-capacitance, $C_m$ (as in usual double QDs[14]). The QD is itself modeled as an Anderson impurity, whose charge is tuned by $\nu$, equivalent to top gate voltage $V_N$. Other top gates ($V_1$, $V_5$) control the couplings of the QD and SI to the source and drain, not included in the model. The output of the model is the energy spectrum of the system, consisting of a few low-lying many-body states and the edge of the continuum. These states are sketched in Fig. 2d between the source and drain. Table 1 shows device and model parameters.

To record the spectrum of excitations between the low-lying many-body states, the device is biased by a source-drain bias

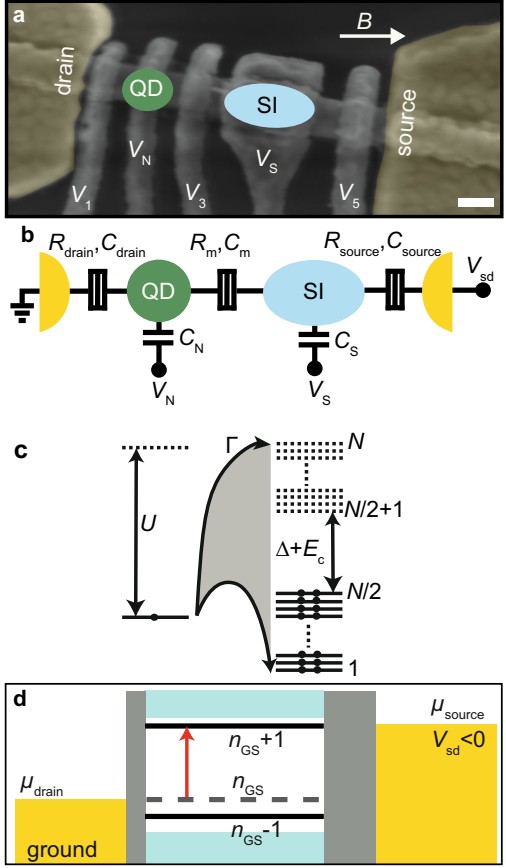

**Fig. 2 Quantum dot-superconducting island device. a** Scanning electron micrograph of the QD-SI device, comprising a top-gated InAs nanowire with an Al SI (underneath top gate S, and therefore not visible) contacted by Ti/Au normal leads. Scale bar = 100 nm. An arrow shows the direction of the applied magnetic field, $B$. **b** Electrostatics of the device. $R$'s and $C$'s denote tunnel resistances and capacitances, while $V$'s indicate voltages. The capacitances and voltages of top gates 1, 3 and 5 are not shown. In a device designed to observe Yu-Shiba-Rusinov subgap excitations, the source would be shorted to the SI. **c** Model of the device. A single QD level is coupled with hybridization strength $\Gamma$ to $N$ levels in the SI. $N$ electrons (shown as dots) fill $N/2$ levels of the SI. In the absence of the QD, it costs an energy $E_c + \Delta$ to excite a quasiparticle in the SI, which is lowered by the coupling to the QD to produce SCE. For odd occupation, the energy cost is instead $E_c - \Delta$. The interdot charging energy, $V$, arising from $C_m$ is not shown. **d** Sketch of how the output of the model is probed in transport. The GS (dashed line) and ±1 excited states (solid lines) corresponding to the parabolas in e.g. Fig. 1e, and the continuum (cyan bands), are coupled by asymmetric tunnel barriers shown as gray rectangles of different width to the source and drain metals (yellow). The barrier asymmetry pins the GS energy to the drain. In this example, the energy difference between the $n_{GS} + 1$ and $n_{GS}$ states matches the bias window, producing a current through the cycle $n_{GS} \rightarrow n_{GS} + 1 \rightarrow n_{GS}$.

Table 1 Parameters of the QD-SI device and model. Top-row parameters are estimates obtained from measurements (for methods, see Supplementary Note 1 and Supplementary Fig. 2), while the bottom-row ones represent best fits of the model output to the experimental data based on the measured parameters as the initial input for subsequent fine tuning.

| $\Gamma$ (meV) | $U$ (meV) | $E_c$ (meV) | $\Delta$ (meV) | $V$ (meV) |
|---|---|---|---|---|
| 0.05 | 0.8 -1.0 | 0.19 | ≤0.27 | 0.13 |
| 0.04 | 0.8 | 0.18 | 0.2 | 0.16 |

while a peak at the opposite polarity that of a many-body state with $n_{GS} - 1$ electrons, where $n_{GS}$ is the total charge in the GS.

The zero-bias $G$ signal exhibits a strong dependence on $V_S$ and $V_N$, as shown in the diagram of Fig. 3a. Singlet ↔ doublet GS transitions are observed in the experiment when conductance lines are crossed, as at $V_{sd} = 0$ these lines appear when the $n_{GS}$ and $n_{GS} + 1$ (or $n_{GS} - 1$) states in Fig. 2d are degenerate at zero energy. The repetition of the central hexagonal charge domain in the $V_S$ direction indicates filling of the SI. As a guide of the filling of the QD and the SI, we approximate their charge expectation values as integers $n_N$, $n_S$ in each of the charge domains. This is an approximation as only $n_{GS}$ is integer with $n_{GS} = n_N + n_S$[13] (see Supplementary Fig. 1). Small but resolvable 1,1 singlet domains (an example is enclosed in a dotted line) are seen between the 1,2 and 1,0 doublet domains. In contrast, the lines to the sides of the central hexagonal domains, which separate the 0,0 and 0,2 domains and the 2,0 and 2,2 domains, show no splitting at this resolution. The difference stems from finite $\Gamma$ and $V$, which stabilize the 1,1 but not the 0,1 and 2,1 domains. The presence of the 1,1 Coulomb-aided YSR singlet is the key difference from a trivial double QD stability diagram[14] and from the $\Delta = 0$ case[9]. For instance, a raise of the interdot coupling in a double QD introduces molecular orbitals which show as avoided crossings at triple points (TPs), whereas in the QD-SI system the YSR singlet is a many-body state for these parameters. Finite $\Gamma$ and $V$ are also responsible for increasing the distance between the points of multiple degeneracy, for the acute angle between vertical and horizontal conductance lines and, in the case of $\Gamma$, for curving the conductance lines.

Our model of the system produces a diagram of GS transitions of the SCE that matches the gate position of the conductance lines, as shown in the comparison of the calculation to the experimental data in Fig. 3a. The quality of the match for model parameters approximately similar to the experimentally measured values (with $\Delta$ as the only fit parameter) constitutes a first proof of the presence of SCE in our device.

We corroborate the spin ($S_z$) assignment done in Fig. 3a (right panel) at $B = 0$ from the variation of GS domain sizes with $B = 0.3$ T (in inset). Doublet domains are stabilized by $B$ more than singlet domains, while triplet domains are stabilized further than doublet and singlet domains. The model fits the data using the $g$-factors as free parameters, and taking into account the GS transition from singlet to triplet in the 1,1 charge sector (charge parabolas are shown in Supplementary Fig. 3). The $g$-factors in the Hamiltonian are significantly larger than the measured effective $g$-factors (see Table 2). These bare $g$ factors produce Zeeman splittings $E_{Z,QD} = g_{0S}\mu_B B$ and $E_{Z,SI} = g_{0N}\mu_B B$ in the QD and the SI, where $\mu_B$ is the Bohr magneton. The effective and bare $g$ factors would be equal if the expectation values of the QD and the SI charges increased in steps of exactly $1e$ across the GS transition lines. For non-zero $\Gamma$, (non-integer) charge distributions occur between the QD and the SI on either side of the GS

voltage, $V_{sd}$, and the differential conductance, $G$, is measured at the grounded drain, as shown in Fig. 2d. Asymmetric source and drain couplings are needed for $G$ to embody the energy asymmetry of the SCE. While we cannot account for the magnitude of $G$, we expect that the measured $G(-V_{sd})$ reflects the excitation energies at $eV_{sd}$, where the negative sign in the argument is necessary to account for the voltage drops and polarity conventions. Symmetric barriers would instead result in a trivial bias symmetry. A peak at one polarity thus demonstrates the existence, at the corresponding excitation energy, of an excited many-body state with $n_{GS} + 1$ electrons in the device,

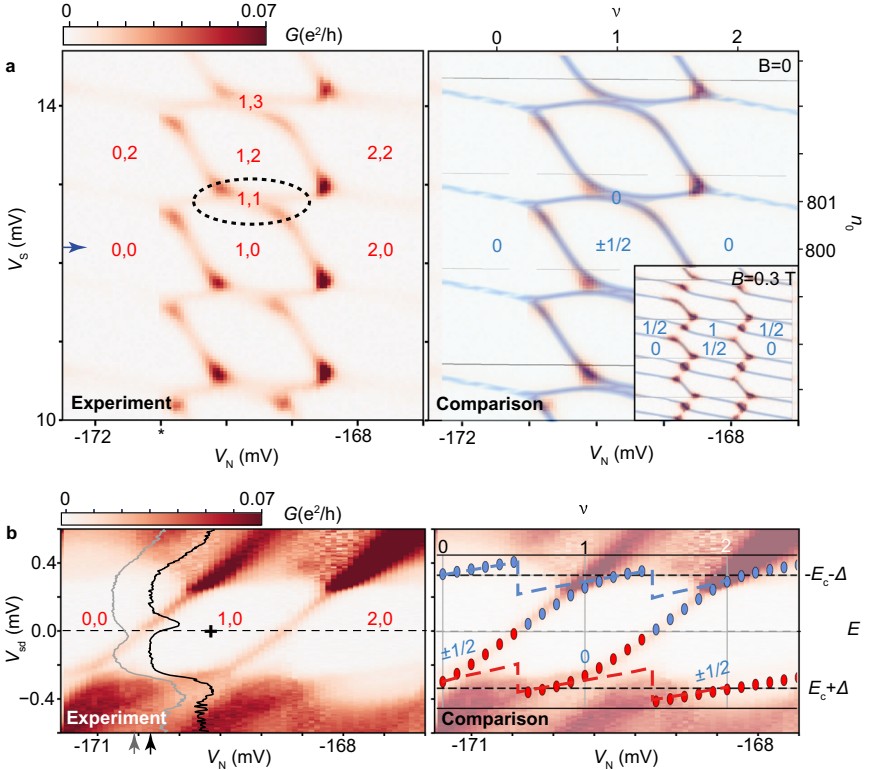

**Fig. 3 Superconducting Coulombic subgap excitations and Coulomb-aided Yu-Shiba-Rusinov singlet. a** Left. Zero-bias differential conductance, $G$, versus SI gate voltage, $V_S$, and QD gate voltage, $V_N$, at magnetic field $B = 0$ measured in the QD-SI device. Other gates are set at $V_1 = -350$ mV, $V_3 = -52$ mV, $V_5 = -169$ mV. An unwanted gate glitch is indicated by an asterisk. The Coulomb-aided YSR singlet domain is encircled. Right. Calculation of GS transitions (blue lines) versus charges induced in the QD, $\nu$, and in the SI, $n_O$, overlaid on a duplicate of the experimental data for $N = 800$ and parameters indicated in Table 1. The graph is a collage of five identical plots with $n_O$ ranging from 799.5 to 801.5. Inset. Zero-bias $G$ versus $V_S$ and $V_N$ at $B = 0.3$ T. The same colorscale and gate settings as the $B = 0$ diagram are used. A calculation of GS transitions (blue lines) versus $\nu$ and $n_O$ is overlaid on the experimental data for $B = 0.3$ T, $N = 200$ and parameters indicated in Tables 1 and 2. **b** Left. Colormap of $G$ versus $V_N$ and source-drain bias voltage, $V_{sd}$, with $V_N$ swept along the blue arrow in **a**. The color scale is saturated to highlight SCE. The overlaid gray and black traces, set to the same $G$ scale and shifted for clarity, are taken at the $V_N$ positions indicated by arrows of the same color. Right. Calculated low-energy SCE spectrum (addition: red, removal: blue dots) for $n_O = 800$ overlaid on the data. The continuum edge is indicated by dashed lines. Approximate QD, SI charges are given in red, while their **a** GS and **b** excitation spin $S_z$ is given in blue.

**Table 2 Effective and bare $g$-factors of the two components of the QD-SI device.** Effective $g$-factors of the QD, $g_N$, and of the SI, $g_S$, extracted from the data in Fig. 3a (for methods, see Supplementary Note 1), and bare $g$-factors $g_{ON}$, $g_{OS}$ used as input in the corresponding finite $B$ calculation.

| $|g_N|$ | $|g_{ON}|$ | $|g_S|$ | $|g_{OS}|$ |
| --- | --- | --- | --- |
| 2.9 | 5 | 6.9 | 20 |

transition line (see Supplementary Fig. 1), hence the effective $g$-factors are some non-trivial function of the true (bare) $g$-factors which appear in the Hamiltonian[15].

Following this comprehensive mapping, we show in Fig. 3b the $G$ spectrum at finite $V_{sd}$ versus $V_N$ for fixed $V_S$, at which the SI contains only Cooper pairs in the GS up to a good approximation. The SCE have a double-S shape, spanning $V_{sd} \approx -0.37 \rightarrow 0.37$ mV. They are approximately inversion symmetric in position and in $G$ intensity with respect to the electron-hole gate-symmetric filling point of the QD, which corresponds to the center of the 1,0 sector (indicated by a cross), from where removing/adding an electron from/to the QD are equally energetically unfavorable. $G$ jumps in intensity when the SCE cross zero bias, as highlighted by

the insert traces at gate points before (gray) and after (black) one of such changes. While the SCE are expected to appear as a pair at asymmetric positive and negative bias positions for a given gate voltage, in practice only one SCE is observed. A GS change brings *discontinuously* up to the continuum the other state, as charge is suddenly redistributed between the QD and the SI[13].

Our model reproduces the position of the subgap resonances, as evidenced in the overlay of the calculated spectrum on the experimental data in Fig. 3b (see also Supplementary Figs. 4–8 and Supplementary Note 2). Differences between the SCE spectrum and the spectrum in the Coulombic ($\Delta = 0$) and Yu-Shiba-Rusinov ($E_c = 0$) limits are shown in Supplementary Fig. 9. The Coulombic spectrum bears resemblance to that of an impurity in the paramagnetic Mott insulator described by the Hubbard model[16,17], despite the differences in the Hamiltonian (local Hubbard interaction versus constant Coulomb repulsion in our model). In both cases the charge transfer from the impurity site to the bath costs energy corresponding to the total charge gap of the system in the absence of the impurity ($\approx U/2$ in the Hubbard model at half-filling, $E_c + \Delta$ in our device), and in both cases there is a (quasi)continuum of fermionic states extending above this gap (doublons/holons in a Mott insulator, and Coulomb quasiparticles with a mixed character of Bogoliubov quasiparticles due to $\Delta$ in our device), leading to the same phenomenology.

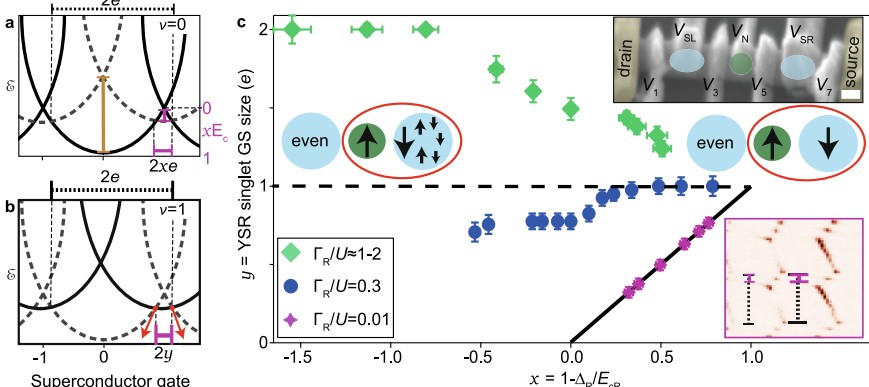

**Fig. 4 From a many-body Yu-Shiba-Rusinov state to a two-body singlet. a, b** Parameter extraction from charge parabolas for **a** an empty QD ($\nu = 0$) and **b** a half-filled QD ($\nu = 1$). For simplicity, the $\Gamma/U \ll 1$ case is illustrated. Dashed (solid) parabolas indicate doublet (singlet) states. **a** For $x > 0$, scaling the horizontal solid bar by the dashed one provides $x = 1 - \Delta/E_c$. The short vertical bar corresponds to $E_c - \Delta$, while the long one equates to $E_c + \Delta$, from which $x$ can be determined when $x < 0$. **b** Scaling the horizontal solid bar by the dashed one provides $y$, a measurement of the YSR singlet ground state stability in units of $e$. $\Gamma$ increases the solid bar size (red arrows). **c** $y$ versus $x$ for different $\Gamma_R/U$ values, measured in a SI-QD-SI device (top inset, scale bar=100 nm) with full tunability of the Coulomb repulsion in the superconductors. The left SI (of even occupation) is used as a cotunnelling probe for the QD-right SI by setting $\Gamma_L/U = 0.02$, $E_{cL}/\Delta_L = 1.5$ (bottom curve), $\Gamma_L/U = 0.3$, $E_{cL}/\Delta_L = 1.6$ (middle curve), or as a soft-gap superconducting probe by setting $\Gamma_L/U \approx 0$, $E_{cL}/\Delta_L \approx 0$ (top curve). The YSR singlet is stabilized by $x$ for weak $\Gamma_R/U$, but it is hindered for large $\Gamma_R/U$. When $x \to 1$, $y$ converges to 1$e$ (dashed line) *independently* of $\Gamma_R/U$, at which point exactly one quasiparticle is bound to the spin of the QD, as sketched in the inset. $y(x = 0)$ provides the YSR binding energy in units of 1$e$. The bottom right inset shows the zero-bias conductance stability diagram for $\Gamma_R/U = 0.01$ at the lowest $x$, to exemplify parameter extraction for $x > 0$. $x$ and $y$ are measured from the thin and thick bars in the diagram, drawn as horizontal bars in **a**, **b**. For $x < 0$, $x$ is extracted from bias spectroscopy by measuring the quantities indicated by vertical bars in **a**. Vertical and horizontal error bars correspond to the sum of full widths at half maximum of the conductance lines delimiting measurement bars for $y$ and $x$. Raw data is shown in Supplementary Fig. 10.

**Dependence of the singlet domain size on the Coulomb repulsion in the superconductor**. To map these limits, we vary continuously the Coulomb repulsion in the superconductor in a second device. We first explore the role of the Coulomb repulsion on the stability of the YSR singlet as the GS. To this aim, we define two quantities, $x = 1 - \Delta/E_c$, and $y$, the YSR singlet GS size in units of $e$. In the $\Gamma/U \ll 1$ regime, $y = (E_c - \Delta + E_B)/E_c$. Fig. 4a, b explain how $x$ and $y$ are experimentally extracted. In the limits when $E_c \to 0$ and $E_c \to \infty$, $x \to -\infty$ and $x \to 1$, respectively. When $E_c = \Delta$, then $x = 0$. Figure 4c shows a measurement of $y$ versus $x$ in a device consisting of a QD coupled to two SIs with hybridization $\Gamma_L$ and $\Gamma_R$ (top inset in Fig. 4c). The SIs have charging energies $E_{cL}$ and $E_{cR}$ and superconducting gaps $\Delta_L$ and $\Delta_R$, and their occupations are tuned with top gate voltages $V_{SL}$ and $V_{SR}$. The advantage of this three-component device over the two-component one is that the presence of only one QD between the two SIs can be verified from stability diagrams similar to that in Fig. 3a against the pairs of gate voltages $(V_N, V_{SL})$ and $(V_N, V_{SR})$. In Fig. 4c, $y$ characterizes the GS stability of the YSR state formed by the binding of the QD spin to the quasiparticle cloud in the right SI, and $x = 1 - \Delta_R/E_{cR}$. To employ the device as this two-component system, the left SI is kept either as a cotunnneling probe at even occupation (for $E_{cL} > \Delta_L$) or as a soft-gap superconducting probe (for $E_{cL} \ll \Delta_L$ and $\Gamma_L \ll \Gamma_R$). Three QD shells with different values of $\Gamma_R/U$ are analyzed. At the weakest $\Gamma_R/U$, $y$ shows a trivial linear dependence with a slope of 1 and with endpoints at (0,0) and (1,1), connected by a fitted solid line in the graph. In this regime, $x$ only *stabilizes* the YSR singlet as the GS. At the other extreme, at the largest $\Gamma_R/U$, $x$ stabilizes the doublet more strongly than the YSR singlet, reducing $y$. In between these two extremes, at $\Gamma_R/U = 0.3$, the behavior is intermediate. When $x \to 1$, $y$ converges to 1$e$ independently of $\Gamma_R/U$, as $x$ stabilizes equally well the doublet and singlet states for even and odd gate-induced charges in the right SI. In the other limit, when $x \to -\infty$, $y$ depends exclusively on $\Gamma_R/U$, as in the usual $E_{cR} = 0$ YSR regime.

**Dependence of the shape of the excitations on the Coulomb repulsion in the superconductor**. Next, we describe how the Coulomb repulsion in the superconductor affects the dispersion of the excitations and how this is related to changes in the stability diagram. In Fig. 5, we show the evolution of the excitations produced by one QD shell on the right SI over a wide range of $V_{SR}$, corresponding to a charge variation of $\approx 960$ electrons. In this range, $E_{cR}/\Delta_R$ goes from 0 to 1.71, as measured from Coulomb-diamonds spectroscopy. The increase in $E_{cR}/\Delta_R$ is reflected on the stability diagram. In the usual $E_{cR} \approx 0$ YSR regime (Fig. 5a), the diagram shows two vertical dispersionless lines, and the spectrum consistently displays a YSR loop (for measurement details, see Methods). When the right SI enters into Coulomb blockade (Fig. 5b, $E_{cR}/\Delta_R = 0.36$), the lines in the stability diagram wiggle as interdot charging and tunneling effects enter into consideration. Consequently, the YSR loop in the spectrum gets skewed rightwards and increases its bias size as the energy gap includes now a Coulombic component. At $E_{cR}/\Delta_R = 0.75$ (Fig. 5c), the entrance of the 1,1 YSR singlet GS breaks the stability diagram into several domains, and the excitations adopt a double-S shape. At this setting, the 1,0 doublet domain has a $V_{SR}$ size $(E_{cR} + \Delta_R + E_{BR})/E_{cR} \approx 2$, where $E_{BR}$ is the YSR binding energy of the spin in the QD to the quasiparticle cloud in the right SI. This results in a maximum of the bias size of the double-S shape excitation. From then on, an increase in $E_{cR}/\Delta_R$ in Fig. 5d–f reduces the energy of the doublet → singlet excitation and the double-S shaped feature shrinks in bias size, concomitantly with the stronger stabilization of the YSR 1,1 singlet GS in the stability diagram.

## Discussion

Throughout this article, we provided compelling evidence for the existence of superconducting Coulombic subgap excitations arising from states bound to a semiconductor-superconductor interface, and we showed how these are related to the usual electron-hole symmetric Yu-Shiba-Rusinov excitations. On one

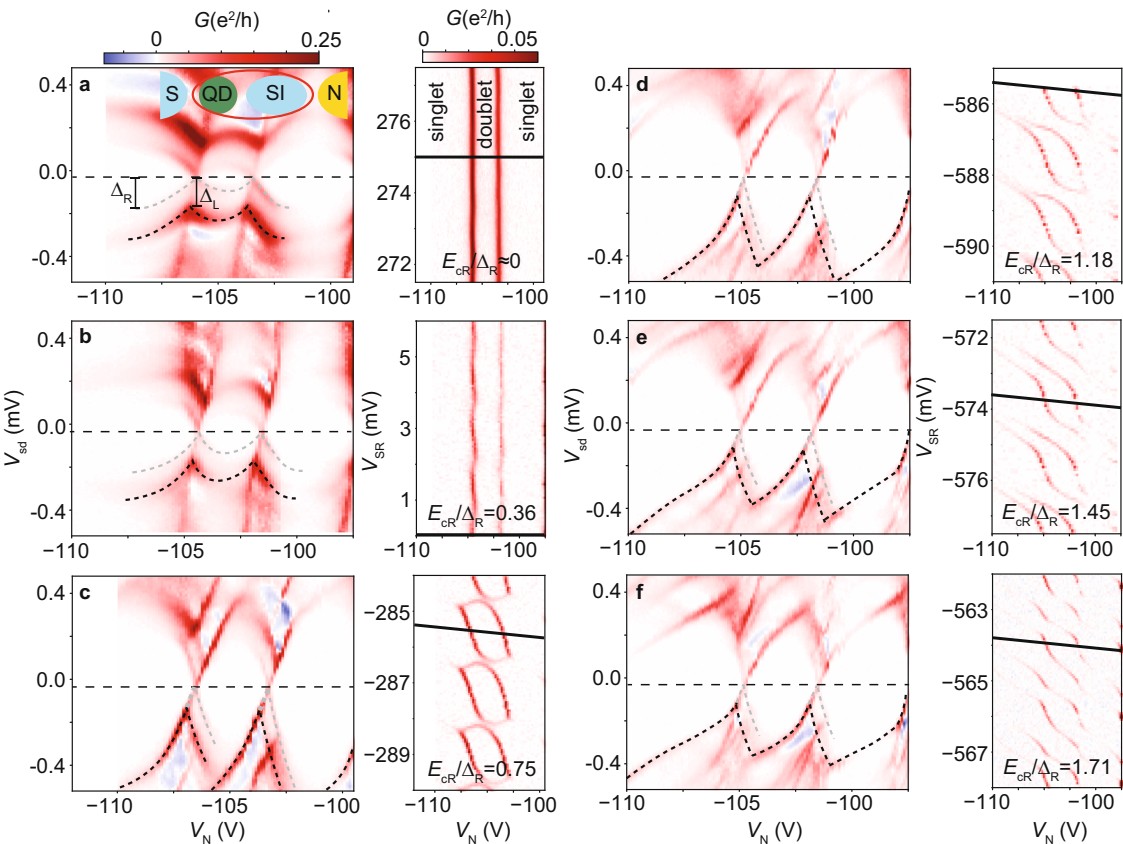

**Fig. 5 From Yu-Shiba-Rusinov to superconducting Coulombic excitations. a–f** Left. Differential conductance, $G$, in the SI-QD-SI device versus source-drain bias voltage, $V_{sd}$, and QD gate voltage, $V_N$. Right. Zero-bias $G$, versus right SI gate voltage, $V_{SR}$, and $V_N$. The measured $E_{cR}/\Delta_R$ ratio is indicated for each pair of panels. For each bias spectrum (left panels), $V_{SR}$ and $V_N$ are swept through the black line on the corresponding stability diagram (right panels), but only $V_N$ is indicated. For ease of comparison, the spectra and diagrams span equal bias and gate relative ranges, and share $G$ scales. Other device parameters are: $U \approx 0.6$–$1.2$ meV, $\Gamma_R/U \approx 0.2$, $\Gamma_L \approx 0$, $E_{cL} = 0$, $\Delta_L = 0.17$ meV. With the left SI tuned to be a soft-gap superconducting probe (as sketched in the inset in (**a**)), black dotted lines highlight the negative-bias side of excitations probed by coherence peaks at $\Delta_L$, while gray dotted lines correspond to replica probed by residual density of states at zero energy. The excitations progressively transform from the characteristic YSR (split) *loop* in **a** at $E_{cR}/\Delta_R \approx 0$ to the *double-S* shape of SCE at $E_{cR} > \Delta_R$ in **b**-**f**. The change is concomitant to the apparition of YSR singlet domains for odd QD occupation in the corresponding stability diagrams. Other gate settings are: $V_{SL} = -50$ mV, $V_3 = -732.5$ mV, $V_5 = -300$ mV, $V_{bg} = -1.49$ V. Right SI parameters are: **a** $E_{cR} \approx 0$, $\Delta_R = 0.15$ meV, **b** $E_{cR} = 0.05$ meV, $\Delta_R = 0.14$ meV, **c** $E_{cR} = 0.12$ meV, $\Delta_R = 0.16$ meV, **d** $E_{cR} = 0.255$ meV, $\Delta_R = 0.215$ meV, **e** $E_{cR} = 0.255$ meV, $\Delta_R = 0.175$ meV, **f** $E_{cR} = 0.24$ meV, $\Delta_R = 0.14$ meV. In the spectra, the true zero bias (indicated by a horizontal dashed line) is offset by $V_{sd} = -18$ $\mu$V, and the stability diagrams are measured at $V_{sd} = -18$ $\mu$V.

hand, we showed that a small Coulomb repulsion in the superconductor is enough to turn the excitations asymmetric in the polarity of the bias voltage. On the other hand, a strong Coulomb repulsion ($E_c \to \infty$) converts the YSR singlet many-body state into a two-body state formed by a spin in the QD and a single quasiparticle in the superconductor.

Though our model is successful at matching excitation energies, an extension which includes transport is needed to account for the magnitude of the conductance features and for their bias positions in devices with more symmetric source-drain barriers. The observation of current blockade in a regime of weaker $\Gamma$ hints at elastic cotunnelling as the transport mechanism in our QD-SI device (see Supplementary Note 2). The absence of zero-bias $G$ in the $\Delta > E_c$ regime (e.g. Supplementary Figs. 10, 11) indicates that Andreev reflection (2$e$ charge transfer) is not a transport mechanism in our devices in this regime. Due to charge transfer between the SI and the QD, the model indicates that an upwards reconsideration of bare $g$-factor values extracted from experimentally-determined $g$-factors of subgap excitations is needed to match the experimental results[6,18]. Based on its success, the model can also inform on future developments, e.g. qubit and

multi-channel devices which utilize the SI-QD-SI device, as outlined in ref. [19].

Given their tunability by gating and by design, our devices can be extended to realize general spin effects, with superconductivity providing an energy gap for resolving the associated excitations. Regular arrays of the demonstrated singlet dimer can be used to generate long-range antiferromagnetic correlations between unpaired spins residing at the end elements[20]. In turn, long-range ferromagnetic correlations can arise in the SI-QD-SI device when made left-right symmetric[21]. Recursive iterations of this device can be used to simulate the intermediate coupling fixed point of the two-channel Kondo model[22]. Schemes of these concepts are shown in Supplementary Fig. 12.

The system sketched in Fig. 1a may also be realized with magnetic adatoms on superconducting droplets (e.g. Pb on an InAs substrate), and probed with scanning tunneling microscopy[4,5]. Several open questions could be answered with this technique: What is the spatial extension of the excitations in a superconducting Coulombic energy gap[23]? Is there orbital structure in the excitations[24]? How do the excitations behave in chains of magnetic adatoms when these chains are deposited on

top of a $E_c > 0$ SI[25]? Do chains of magnetic adatoms deposited on finite $E_c$ SIs support Majorana excitations[26]?

## Methods

**Devices fabrication and layout.** *QD-SI device* (Fig. 2a). A 110-nm wide InAs nanowire picked with a micromanipulator was contacted by 5/200 nm Ti/Au (in yellow) source and drain leads. The ≈350-nm long, 7-nm thick epitaxially-grown Al SI covering three facets of the nanowire was defined by chemically etching the upper and lower sections of the nanowire before contacting. After insulating the nanowire and the leads with a 6-nm thick film of HfO₂, five Ti/Au top gates were deposited along the nanowire. The QD was defined in the bare nanowire next to the SI by setting top gates 1 and 3 to negative voltage. A Si/SiO₂ substrate backgate was kept at zero voltage throughout the experiment.

*SI-QD-SI device* (Supplementary Fig. 10). The SI-QD-SI device was fabricated using a nanowire from the same growth batch. Two nominally identical 7-nm thick, ≈300-nm long, epitaxially-grown Al SIs were defined by chemical etching. The nanowire was contacted by 5/200 nm Ti/Au leads, and then insulated by a 5-nm thick layer of HfO₂ from seven Ti/Au top gates deposited after. The QD was defined between the two SIs by setting top gates 3 and 5 to negative values. The substrate backgate was used to aid the top gates in depleting the device. $E_{cR}/\Delta_R$ was tuned by using an auxiliary QD ($QD^R_{aux}$) defined between the right SI and the source lead. When $QD^R_{aux}$ was put near resonance by sweeping $V_{SR}$, $E_{cR} - \Delta_R$ could be tuned to negative values, and when $QD^R_{aux}$ was put in cotunnelling, $E_{cR} - \Delta_R$ could be tuned to positive values. Similarly, $E_{cL} - \Delta_L$ was tuned using an auxiliary QD defined between the left SI and the drain lead.

The critical $B$ of the superconducting Al film was measured to be $B_c = 2.1$ T in nanowire devices made from the same batch of nanowires used in the fabrication of the present device[27,28], which left ample room for $B$-resolved measurements in the superconducting state. In the QD-SI and SI-QD-SI devices, the presence of superconductivity at large $B$ was determined from size differences of adjacent charge domains with odd and even occupation of the SI, observed up to $B = 1.2$ T and $B = 1.5$ T, respectively. Larger $B$ was not explored.

**Differential conductance measurements.** A standard lock-in technique was used to measure the differential conductance, $G = dI/dV_{sd}$, of the QD-SI device by biasing the source with an AC excitation of 5 μV at a frequency of 223 Hz on top of a DC source-drain bias voltage, $V_{sd}$, and recording the resulting AC and DC currents on the grounded drain lead. In the case of the SI-QD-SI device, $G$ was measured at the grounded drain with a 5 μV lock-in excitation applied at the source at 84.29 Hz. The measurements were performed in an Oxford Triton dilution refrigerator at 30 mK for the QD-SI device and 35 mK for the SI-QD-SI device, such that $k_BT \ll E_c$, where $k_B$ is the Boltzmann constant and $T$ is the refrigerator temperature.

**Calculation of subgap and continuum excitations.** The calculations were done using the density-matrix renormalization group approach (details in Supplementary Note 3). The quantum numbers are the total number of electrons in the system, $n$, the $z$-component of the total spin, $S_z$ (see Supplementary Fig. 3), and the index for states in a given $(n, S_z)$ sector, $i = 0, 1, \dots$. The superconducting Coulombic excitation energies are given by $E = \mathcal{E}(n = n_{GS} \pm 1, S_z = S_{z,GS} \pm 1/2, \mathbf{i} = \mathbf{0}) - \mathcal{E}(n = n_{GS}, S_z = S_{z,GS}, i = 0)$. The edges of the continuum excitations are given by $E_{edge} = \mathcal{E}(n = n_{GS} \pm 1, S_z = S_{z,GS} \pm 1/2, \mathbf{i} = \mathbf{1}) - \mathcal{E}(n = n_{GS}, S_z = S_{z,GS}, i = 0)$. Due to the finite size of the SI, the continuum is in truth only a quasi-continuum of states. The nature of these states and the excitation energies depend on the values of $\Delta$ and $E_c$. For $\Delta = 0$, the quasiparticles are free-electron states. For $\Delta \neq 0$, these are Bogoliubov quasiparticles with pronounced inter-level pairing correlations $\langle c^\dagger_{i\uparrow} c^\dagger_{i\downarrow} c_{j\downarrow} c_{j\uparrow} \rangle$. If $E_c = 0$, the excitation spectrum is not affected by the number of preexisting particles in the superconductor (up to finite-size effects). If $E_c \neq 0$, the particle-addition and particle-removal energies are affected by the charge repulsion (parabolas). The calculations do not provide direct results for the differential conductance of the system, only information about the energies of the GS and the low-lying excitations.

**Spectral measurements in Fig. 5.** To obtain sharp spectral features visible over the continuum background, we tuned the left SI into a superconducting probe ($E_{cL} = 0$, $\Gamma_L \approx 0$). The strong hybridization of the left SI with the drain needed to achieve $E_{cL} = 0$ resulted in an unintended soft gap in this probe, which produced faint replica of the main excitations. For example, in Fig. 5a, black dotted lines correspond to the YSR loop coming from the QD-right SI being probed by the coherence peaks of the probe. The loop is thus followed by negative differential conductance (NDC) and appears at $\mp eV_{sd} = \pm \Delta_L + \mathcal{E}(n = n_{GS} \pm 1, S_z = S_{z,GS} \pm 1/2, \mathbf{i} = \mathbf{0}) - \mathcal{E}(n = n_{GS}, S_z = S_{z,GS}, i = 0)$, reaching $\Delta_L$ at GS transitions. The gray dotted lines highlight a YSR replica probed by the soft gap of the probe, thus an order of magnitude weaker in conductance and without associated NDC. This replica appears at $\mp eV_{sd} = \mathcal{E}(n = n_{GS} \pm 1, S_z = S_{z,GS} \pm 1/2, \mathbf{i} = \mathbf{0}) - \mathcal{E}(n = n_{GS}, S_z = S_{z,GS}, i = 0)$, and therefore crosses zero bias at GS transitions. When $E_{cR}/\Delta_R$ increases, the relationships between the excitations and the bias positions of the conductance features become approximations due to non-ideal

$\Gamma_{source}$, $\Gamma_{drain}$ asymmetry in the device, which also results in negative-slope features included in the dotted lines but not accounted by the model, as the model does not consider transport. The interpretation of features in the continuum at higher bias is outside the scope of this work.

## Data availability

The experimental data generated in this study have been deposited in the ERDA database of the University of Copenhagen at https://doi.org/10.17894/ucph.58ab2544-e746-47d9-a241-b2f19d595b1c.

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

## Acknowledgements

We thank Jens Paaske and Silvano De Franceschi for useful discussions. J.C.E.S. acknowledges funding from the European Union's Horizon 2020 research and innovation program under the Marie Sklodowska-Curie grant agreement No. 832645, and support from the Novo Nordisk Foundation project SolidQ. J.N. acknowledges funding from the FET Open AndQC, the Novo Nordisk Foundation project SolidQ, the Carlsberg Foundation, the Independent Research Fund Denmark, QuantERA 'SuperTop' (NN 127900), the Danish National Research Foundation, and Villum Foundation project No. 25310. A.V. and K.G.R. acknowledge funding from the Sino-Danish Center. P.K. acknowledges support from Microsoft and the ERC starting Grant No. 716655 under the Horizon 2020 program. L.P. and R.Ž. acknowledge the support from the Slovenian Research Agency (ARRS) under Grant No. P1-0044 and J1-3008.

## Author contributions

J.C.E.S. and A.V. performed the experiments. P.K. and J.N. developed the nanowires. J.C.E.S., A.V., K.G.-R., J.N., L.P. and R.Ž. interpreted the experimental data. L.P. and R.Ž. did the theoretical analysis. J.C.E.S. wrote the manuscript with input from A.V., K.G.-R., J.N., L.P. and R.Ž.

## Competing interests

The authors declare no competing interests.
