## [Peer Review File · Nature Communications]

REVIEWER COMMENTS

Reviewer #1 (Remarks to the Author):

The manuscript "Superconducting Coulombic subgap states" by Saldaña et al. reports on transport-spectroscopy experiments in a device composed of a quantum dot (QD) and of a superconducting island (SI). The complex patterns of the measured stability diagrams are well reproduced by a model which accounts for the presence of charging effects both on the QD and in the SI, and of tunneling and capacitive charging between the systems. The data are very interesting and surely worth of publication. Their interpretation however, and most of all the way the paper is written, is highly unclear and extremely confusing. I cannot recommend publication of the work as it is. I urge the authors to eliminate superfluous paragraphs in the main text and elaborate better on the main findings of the work. Likewise, a selection should be done for the extended data and the Supplementary Material. I elaborate on these criticisms and possible improvements below.

i) The coupled QD-SI system bears strong similarities with a double quantum dot setup, whose physics is by now well understood. The main difference here is that only the QD displays both charge and size quantisation because in the superconducting island only Coulomb interaction effects, responsible for superconductivity and charging, are relevant. Indeed, the authors often refer to the DQD for modeling or comparison of the transport spectroscopy data. In this context the sketch in Fig. 1a is rather confusing, since for example the Kondo effect needs a strong hybridization to a large reservoir which is not the case presented here. Similar considerations apply to the Shiba states in Fig. 1b. On the hand, as stated by the authors, the coupling is weak or at most intermediate (see also point ii). On the other hand it is even arguable if it is legitimate to call the SI a "reservoir", as done by the authors. According to the orthodox theory of Coulomb blockade, the occupation probability of the island with a given number of charges is a dynamical quantity. It would be helpful to clarify these points and discuss the physics of DQDs and the difference with the present set-up in more detail.

ii) In the supplemental material (SM) it is argued that one is in an "intermediate" coupling regime; still from Table I the hybridization Γ is the smallest energy scale. Is the tunneling dynamics sequential or are cotunneling-like processes relevant? Which are the signatures? It would be very helpful to clarify this point already at the level of the main text. If it possible to attribute charge and spin states individually, like in Fig. 3, one would infer that the tunneling is still a perturbation.

iii) If the two systems are strongly coupled, both through the hybridization as well as electrostatically, why should a "simple Bardeen tunneling approximation, with the differential conductance being proportional to the QD spectral function, be sufficient to capture the current-voltage characteristics?

The system is composed by a QD and a SI in series, whereby the differential conductance should be more complicated. For example, also the spectral function of the SI should enter. On top of this, the presence of a QD-SI exchange coupling shows that this is a complex many-body system and thus a separation of the two subsystems could be arguable.

iv) Central to the work is the authors' claim that they "provide the first spectral evidence of the subgap states in a superconducting/Coulombic gap". It is of utmost importance to clarify this point. In the transport data one is looking at resonances at the level of electron chemical potentials for different particle numbers, as is clear from the numbers apposed to the data in Fig. 3b and 3c. This is a complex many-body problem. So what is the relation between the measured resonances and the "subgap states"; where are these "states" located? It can be also helpful to better explain what is the ground state transition one is

referring to with the blue lines in Fig. 3b.

v) The sketch in Fig. 3a is also confusing. What one would like to transmit is the information about the addition spectrum of the superconductor. Here, again, chemical potentials yield the proper physical picture. This is not what the filled levels in the sketch convey. As discussed in the caption they rather serve to elucidate the theoretical model where a hybridization term to $N/2$ single particle levels occupied with N electrons is assumed. This is confusing because due to the many-body pairing interaction, the spectrum looks pretty different. Part of the N electrons will form Cooper pairs and part of them will occupy quasiparticle states. When an electron is added to the SI, a whole rearrangement will take place. Nevertheless the addition spectrum will still exhibit a gap of the order of $\Delta + E_c$, as suggested by the sketch. If two electrons are added, the situation might be different, since a Cooper pair might form. What is then the picture the reader should have in mind?

vi) The authors often refer to Majorana physics to give a motivation to their work. Several sentences and long paragraphs are used to this scope. While some motivation can be put in introduction and conclusion, I would suggest to leave out long paragraphs from the main part of the manuscript. That space should be used to clarify the novel physics of this interesting QD-SI system.

For example starting from line 71: "Motivated by the use of SIs in the search for Majorana states, we also investigate the B dependence of the subgap states". I find the B dependence very interesting per se, without the need to invoke Majorana physics or topological superconducting island.

Likewise there is a very long paragraph starting from line 191 and ending on line 209 which should be shortened: "Our experiment includes a tuning knob for the filling of the SI, VS, which is a feature in common with experiments using a SI on a semiconductor nanowire to spot Majorana states. In these experiments, B opens a topological superconducting gap in the nanowire segment under the SI. S_e and S_o are expected to oscillate around $1e$ due to overlap of Majorana states."

Also the sentence starting from line 220 distracts the reader from the main focus of the paper: "Being the first spectroscopy of the subgap excitations of a QD-SI nanowire device at finite B, our measurement is an important preamble for parity readout of topological qubits using QDs coupled to topological SIs in semiconductor nanowires."

Reviewer #2 (Remarks to the Author):

In this paper, the authors implemented spectroscopy of subgap states in nanowire-superconductor hybrid structures. In particular, they detected and characterized in detail the subgap structure when the superconducting island has charging energy, unlike the Andreev bound or YSR states that have been studied extensively.

The device structure is expected to be a platform for detecting and controlling the Majorana Fermions, which is currently attracting much attention, and the authors mention the relevance of the Majorana Fermions in their discussion of the magnetic field dependence. It is interesting to note that the number of charges can change by one for a change in V_n even though the superconducting islands are connected in series.

The paper provides useful and valuable information for those who study the physics of superconductor-semiconductor junctions.

However, in my opinion, the results do not have the necessary impact to be published on

Nature Communications. The YSR and Andreev states appear not only in such superconducting semiconductor junctions, but also in other research areas like the surface physics. However, the physics discussed in this paper does not seem to have the same interdisciplinary scope and general interest as the Andreev and YSR states. Therefore, I do not recommend this paper for Nature Communications; I think it would be good for Communications Physics.

I have the following comments:

1. As for Figure 1, I think that the figures a~d are not necessary or not suitable for the explanation of the superconducting Coulombic states. In this paper, the authors mostly compare with YSR, and I think Extended Data Figure 1 is more helpful for understanding than Figure 1.
2. In Fig. 3b, the change in charge state with respect to V_n is assumed to be one by one. Can this be confirmed from the experimental data? (Is this validated by FigS2 because there is no S_0 expansion in the V_n dependence?)
3. In Figure 4, there is a description of the triplet superconducting Coulombic state, but I think it is unclear whether the superconducting island remains "superconductive" when $B=0.35$ T or higher. Therefore, it may be difficult to discuss the phase in $B>0.35$ T as a triplet superconducting Coulombic state. The data obtained at $B=0.35$ T in Extended Data Figure 7 seems to be very similar to that of Coulomb diamond in the normal state. In addition, (as the authors mention in the manuscript) S_0 expansion above 0.35 T cannot be found in Fig. 4. If the authors cannot show experimentally that the superconductivity is still there, I think that this argument is too strong.
4. The device configuration is very similar to the previous Majorana signature report (10.1126/science.aaf3961, <https://doi.org/10.1038/nature17162>). In the discussion part, the authors may compare the experimental results with/without the QD or the charging energy in the superconducting island. Why do the authors not indicate such spectroscopic results in the high magnetic field regime? If there are some reasons why the authors cannot obtain the Majorana signature in this device, it is helpful to write them on the manuscript.

Reviewer #3 (Remarks to the Author):

The joint experimental and theoretical work focuses on the subgap spectrum of a quantum dot attached to a superconductor island with finite size and where the superconducting gap and the charging energy of the island are comparable. Extensive transport data is presented and there is an impressive agreement between experimental findings and theory. The topic is timely, since superconducting subgap states are at the center of attention in the research of hybrid devices due to their potential in novel qubit. The work is original, based on my knowledge, subgap states have not been studied in such geometry.

The presentation is clear, however it is over concise, the reader should rely on Extended Data Figures to follow the manuscript. Especially content from the first three Extended Data Figures are missing.

Altogether I find the work valuable to be publish in Nature Communications if the following points are properly addressed:

- Superconducting islands with finite charging energy was addressed in the past e.g. in metallic grains e.g. by D.C. Ralph et al. Short review of these works is missing in the introduction either on the experimental or on the theory side.

- Based on Fig. 1c and on standard double dot physics, it is a rather straight forward expectation to receive an asymmetric spectrum of SCS for $E_C > 0$. Reading through the conclusion of the paper, the reader is left with a sense of lack, what motivated the study of this particular geometry? It would be desirable to extend the end of the manuscript with a section how this setup could be used for e.g. qubits.
- Spin-orbit interaction generates a non trivial spin character of electron states. When electrons tunnel from the supra island to the dot this generates a non-spin conserving tunneling between the two natural spin basis. How that would effect the spectrum, is there some experimental signature of this effect? Does it influence the presented g-factor study?
- For completeness please add other parameters e.g. V , Γ in Fig. 2b.
- Does the reader understand well that the Al superconductor is epitaxial and grown in MBE after wire growth?
- It is stated that at B_D in Fig. 4 3 states get degenerate. Please explain clearer in the text how more than 3 states could be degenerate at a particular point. In general it is unlikely.
- For Extended Data Figure 3b the caption seems not precise.
- In Fig. 3b (experiment) FP point has larger conductance as TP. Does it consistent with theoretical expectation?

Reviewer #1 (Remarks to the Author):

The manuscript “Superconducting Coulombic subgap states” by Saldaña et al. reports on transport-spectroscopy experiments in a device composed of a quantum dot (QD) and of a superconducting island (SI). The complex patterns of the measured stability diagrams are well reproduced by a model which accounts for the presence of charging effects both on the QD and in the SI, and of tunneling and capacitive charging between the systems. The data are very interesting and surely worth of publication.

We thank Reviewer #1 for finding our data interesting and for acknowledging the good agreement with our model of the device.

Their interpretation however, and most of all the way the paper is written, is highly unclear and extremely confusing. I cannot recommend publication of the work as it is. I urge the authors to eliminate superfluous paragraphs in the main text and elaborate better on the main findings of the work. Likewise, a selection should be done for the extended data and the Supplementary Material.

I elaborate on these criticisms and possible improvements below.

We thank the reviewer for his/her detailed comments. We have now made our manuscript clearer and more concise following these suggestions.

Elaboration on the main findings of the work:

To elaborate on the two main findings of the work indicated in the abstract, we have added two figures (new Figs. 4 and 5) measured in a second device showing the effect of the Coulomb repulsion in the superconducting island on the spectrum and on the shape of the stability diagram.

Improving clarity:

1. The introduction has been shortened to two paragraphs which present the most proximate context, which is YSR excitations and superconducting islands with finite E_c .
2. New Fig. 1a depicts in the most general way the system under consideration: a spin impurity coupled to a finite E_c superconducting grain. This invites further experimental work in realizations of this system beyond our QD-SI and SI-QD-SI devices.
3. New Figs. 1b,c,d,e compare the YSR charge parabolas ($E_c=0$) to the $E_c>0$ charge parabolas. Two core messages of the paper are introduced here: a) E_c stabilizes the singlet state for $n_0=1$, $\nu=1$, and b) it makes the excitations between doublet and singlet states electron-hole asymmetric.
4. New Fig. 2 groups in one figure the device image and sketches of its electrostatics, the model, and how the experiment probes the output of the model, with color conventions providing direct links between Fig. 1d, Fig. 2d and Fig. 3b.
5. The B dependence data has been removed to avoid diluting the main message of the paper. Only a stability diagram at $B=0.3$ T is shown in Fig. 3a to support the assignment of the spin of the states.
6. With the same aim, the discussion of multi-degeneracy points has been removed from the main text.

Superfluous paragraphs removed:

1. The Kondo effect is now not mentioned in the introduction to avoid confusion as the Kondo effect is not probed in the experiment.

2. Paragraphs/sentences related to Majorana zero modes which interrupted the flow of the presentation of the results have now been removed from the text. The relevance of our work for this field of research is only briefly mentioned in the introduction.

Selection done on the Supplementary Material:

1. All B dependence data has been removed. Consequently, the discussion of Majorana modes has also been removed.
2. Supplementary Figures showing gate tuning of Γ and Γ_{source} have been removed.
3. New Supplementary Figs. 3 and 4 have been reduced in number of panels.

Selection done on the Supplementary Material:

1. Extended Data Fig. 1 has been removed and instead Figs. 1b,c,d,e take its place.
2. Extended Data Figures showing B-dependence have been removed except for new Extended Data Fig. 2.
3. The Extended Data Figure related to the SI-QD-SI device has been replaced by new Extended Data Figure 5.

Other changes to improve clarity are provided in our detailed response to Reviewer #1's comments below.

i) The coupled QD-SI system bears strong similarities with a double quantum dot setup, whose physics is by now well understood. The main difference here is that only the QD displays both charge and size quantisation because in the superconducting island only Coulomb interaction effects, responsible for superconductivity and charging, are relevant. Indeed, the authors often refer to the DQD for modeling or comparison of the transport spectroscopy data.

Indeed, we compare the measured stability diagrams to those from the double QD setup because they are familiar to most readers. We now make the differences between the two systems explicit on p. 3, col. 2: "The presence of the 1,1 Coulomb-aided YSR singlet (..) is a many-body state for these parameters."

In this context the sketch in Fig. 1a is rather confusing, since for example the Kondo effect needs a strong hybridization to a large reservoir which is not the case presented here. Similar considerations apply to the Shiba states in Fig. 1b. On the hand, as stated by the authors, the coupling is weak or at most intermediate (see also point ii).

We have removed explicit mentions of the Kondo effect from the paper to avoid confusion, as we do not treat this effect experimentally.

However, the statement made by the referee with regards to YSR states is incorrect: strong hybridization to a large reservoir is not the only case in which YSR states can appear. Our work shows that a Coulomb-aided YSR singlet emerges in the 1,1 charge sector as a result of E_c in the SI. This is the case even for arbitrary small hybridization, as long as $E_c > \Delta$.

Using a second device in which E_c could be tuned with a gate voltage, we directly observe the (linear) stabilization of the YSR singlet with E_c for weak hybridization, and its destabilization with E_c for strong hybridization. We gather our observations in new Fig. 4, based on the data shown in new Extended Data Fig. 5.

Not only we observe these two novel effects of E_c on the YSR singlet stability, but we also show experimentally on new Fig. 4 that the nature of the YSR singlet changes from a many-body state to a two-body state with exactly one quasiparticle in the SI for strong Coulomb

repulsion in the SI. This occurs independently of the hybridization between the QD and the SI.

On the other hand it is even arguable if it is legitimate to call the SI a “reservoir”, as done by the authors.

If “reservoir” is taken to mean something that can provide electrons at zero cost, then indeed the SI is not. The SIs are strictly reservoirs only when they are heavily hybridized with the metallic leads such that $E_{cL}=E_{cR}=0$, e.g. in new Fig. 5a. We now avoid the use of this term.

According to the orthodox theory of Coulomb blockade, the occupation probability of the island with a given number of charges is a dynamical quantity.

True.

It would be helpful to clarify these points and discuss the physics of DQDs and the difference with the present set-up in more detail.

This is now done as mentioned above.

ii) In the supplemental material (SM) it is argued that one is in an “intermediate” coupling regime; still from Table I the hybridization Γ is the smallest energy scale.

We now include a classification of the three different Γ regimes in the SM under “Other Γ regimes in the QD-SI device” based on experimental considerations in the stability diagram, which has the strength of being model-independent.

Is the tunneling dynamics sequential or are cotunneling-like processes relevant? Which are the signatures? It would be very helpful to clarify this point already at the level of the main text.

This point is now discussed on p. 7: “The observation of current blockade in a regime of weaker Γ , detailed in the Supplementary Information, hints at elastic cotunnelling as the transport mechanism in our QD-SI device. The absence of zero-bias \mathbb{G} in the $\Delta > E_{\text{c}}$ regime (e.g. Extended Data Fig. 5) indicates that Andreev reflection ($2e$ charge transfer) is not a transport mechanism in our devices in this regime.”

However, we stress that the metallic leads do not appear in the Hamiltonian of our model, and the differential conductance is not calculated, which would be needed to fully determine the transport mechanism by comparison to the experimental data.

If possible to attribute charge and spin states individually, like in Fig. 3, one would infer that the tunneling is still a perturbation.

Reviewer #1 is referring to the charge and spin states attributed to the different GS sectors of the stability diagram and subgap spectrum. As we explain in the main text, it is not possible to attribute charge or spin states individually to the QD and SI. The charges n_N and n_S attributed individually to the QD and the SI in the various stability diagrams and subgap spectra are obtained by rounding the calculated (fractional) expectation values of the charge in each of the two parts of the system. Only the total charge n_N+n_S is an integer value. We show calculations of how the expectation values evolve with gate voltages in Extended Data Fig. 1. Calculations of the gate dependence of the expectation value of S^2 of the QD are shown in Fig. 5 of Pavešić, L., Bauernfeind, D., & Žitko, R. (2021). Subgap states in superconducting islands. Phys. Rev. B, 104(24), L241409.

The tunnelling between the QD and the SI is not a perturbation, as it produces many-body states.

iii) If the two systems are strongly coupled, both through the hybridization as well as electrostatically, why should a “simple Bardeen tunneling approximation, with the differential conductance being proportional to the QD spectral function, be sufficient to capture the current-voltage characteristics? The system is composed by a QD and a SI in series, whereby the differential conductance should be more complicated. For example, also the spectral function of the SI should enter. On top of this, the presence of a QD-SI exchange coupling shows that this is a complex many-body system and thus a separation of the two subsystems could be arguable.

We have removed this sentence from the paper.

The Bardeen mechanism is indeed strictly speaking incorrect in our devices due to sizable Γ . For asymmetric Γ_{source} and Γ_{drain} , invoking this mechanism is not necessary to relate excitation energies to features in the measured differential conductance against V_{sd} , no matter what the details are (the details are about the amplitude of the observed features).

A separation of the two subsystems (either in charge or spin) is strictly incorrect as explained in the previous point, but we believe that our cartoons of the system in which these are separated are pedagogical for the reader who comes from the QD community.

iv) Central to the work is the authors’ claim that they “provide the first spectral evidence of the subgap states in a superconducting/Coulombic gap”. It is of utmost importance to clarify this point. In the transport data one is looking at resonances at the level of electron chemical potentials for different particle numbers, as is clear from the numbers opposed to the data in Fig. 3b and 3c.

This is a complex many-body problem. So what is the relation between the measured resonances and the “subgap states”; where are these “states” located?

We have corrected the terminology, including a slight change in the title of the paper, which also reflects that for fine-tuned parameters electron-hole symmetric Yu-Shiba-Rusinov excitations can exist in our device even when $E_c > 0$. The term “subgap states” has been replaced by “excitations”, corresponding to the energy difference between the ground state and the first excited state (differing by one particle). This energy can lie below the energy gap, and therefore these are “subgap excitations”.

The states are located at the semiconductor/superconductor interface where the QD and SI meet. This is evidenced by the effect of the gate located between the QD and SI plunger gates on the shape of the stability diagram. However, a STM technique is needed to have a more precise answer, as mentioned in the last paragraph of the main text.

It can be also helpful to better explain what is the ground state transition one is referring to with the blue lines in Fig. 3b.

To make this clearer, we have added on p. 4, col. 1 the phrase: “Singlet \leftrightarrow doublet GS transitions are observed in the experiment when conductance lines are crossed, as at $V_{\text{sd}}=0$ these lines appear when the n_{GS} and $n_{\text{GS}}+1$ (or $n_{\text{GS}}-1$) states in Fig. 2d are degenerate at zero energy.”

v) The sketch in Fig. 3a is also confusing. What one would like to transmit is the information about the addition spectrum of the superconductor. Here, again, chemical potentials yield the proper physical picture. This is not what the filled levels in the sketch convey. As discussed in the caption they rather serve to elucidate the theoretical model where a hybridization term to $N/2$ single particle levels occupied with N electrons is assumed. This is confusing because due to the many-body pairing interaction, the spectrum looks pretty different. Part of the N electrons will form Cooper pairs and part of them will occupy quasiparticle states. When an electron is added to the SI, a whole rearrangement will take place. Nevertheless the addition spectrum will still exhibit a gap of the order of $\Delta + E_c$, as suggested by the sketch. If two electrons are added, the situation might be different, since a Cooper pair might form. What is then the picture the reader should have in mind?

We have simplified and grouped our schematics on Figs. 1 and 2 as follows:

- 1) Fig. 1a. The idealized system, which could be any of the following: a magnetic atom/molecule adsorbed to a superconducting grain or droplet, or a QD coupled to a SI.
- 2) Fig. 2b. The electrostatics of the device.
- 3) Fig. 2c. The model of the QD-SI system for the case of even occupation in the SI. We now clarify in the caption that for odd occupation in the SI the energy addition cost is $E_c - \Delta$. Note that there are no leads in this sketch (as it is the case for our model), so that the reader does not confuse this sketch as indicative of transport.
- 4) Fig. 2d. Transport through the system. The diagonalized excited many-body states are sketched with asymmetric energies with respect to the ground state energy, sandwiched between the two leads of the device. Note that a particular transport mechanism is not assumed. The notion of chemical potentials of the leads aligned with the energy difference between the excited and ground states is present in the scheme.

vi) The authors often refer to Majorana physics to give a motivation to their work. Several sentences and long paragraphs are used to this scope. While some motivation can be put in introduction and conclusion, I would suggest to leave out long paragraphs from the main part of the manuscript. That space should be used to clarify the novel physics of this interesting QD-SI system.

These paragraphs have been removed, with a brief motivation included only in the introduction.

For example starting from line 71: "Motivated by the use of SIs in the search for Majorana states, we also investigate the B dependence of the subgap states". I find the B dependence very interesting per se, without the need to invoke Majorana physics or topological superconducting island.

We have removed this line. We have also largely removed the B dependence because, as interesting as it may be, it diluted the main message of our manuscript and resulted in extensive use of the Supplementary Information and Extended Data Figures. We only keep an essential stability diagram in Fig. 3 at $B=0.3$ T for supporting our assignment of the spin of the charge domains.

Likewise there is a very long paragraph starting from line 191 and ending on line 209 which should be shortened: "Our experiment includes a tuning knob for the filling of the SI, V_S , which is a feature in common with experiments using a SI on a semiconductor nanowire to spot Majorana states. In these experiments, B opens a topological superconducting gap in

the nanowire segment under the SI. S_e and S_o are expected to oscillate around $1e$ due to overlap of Majorana states.”

This paragraph has been removed.

Also the sentence starting from line 220 distracts the reader from the main focus of the paper: “Being the first spectroscopy of the subgap excitations of a QD-SI nanowire device at finite B , our measurement is an important preamble for parity readout of topological qubits using QDs coupled to topological SIs in semiconductor nanowires.”

This sentence has been removed.

Reviewer #2 (Remarks to the Author):

In this paper, the authors implemented spectroscopy of subgap states in nanowire-superconductor hybrid structures. In particular, they detected and characterized in detail the subgap structure when the superconducting island has charging energy, unlike the Andreev bound or YSR states that have been studied extensively.

The device structure is expected to be a platform for detecting and controlling the Majorana Fermions, which is currently attracting much attention, and the authors mention the relevance of the Majorana Fermions in their discussion of the magnetic field dependence. It is interesting to note that the number of charges can change by one for a change in V_n even though the superconducting islands are connected in series.

The paper provides useful and valuable information for those who study the physics of superconductor-semiconductor junctions.

However, in my opinion, the results do not have the necessary impact to be published on Nature Communications. The YSR and Andreev states appear not only in such superconducting semiconductor junctions, but also in other research areas like the surface physics.

However, the physics discussed in this paper does not seem to have the same interdisciplinary scope and general interest as the Andreev and YSR states.

Therefore, I do not recommend this paper for Nature Communications; I think it would be good for Communications Physics.

We thank Reviewer #2 for acknowledging that our paper contains valuable information for those who study the physics of superconductor-semiconductor junctions, and for his/her detailed comments, to which replies are formulated below.

However, we first address some inexactitudes in the general assessment of our work made above.

Reviewer #2 states: “They detected and characterized in detail the subgap structure when the superconducting island has charging energy, **unlike** the Andreev bound or YSR states that have been studied extensively.” This is incorrect. Having $E_c > 0$ in the SI only increases the parameter space in which YSR and Andreev states exist; it does not forbid the formation of these other states. If anything, our devices constitute a **generalization** of these states; the usual YSR and Andreev state physics studied in previous devices occurs in the tiny parameter region corresponding to $E_c = 0$. Our second device demonstrates this directly in Fig. 5a. For $E_c > 0$, however, the region of existence of YSR excitations is restricted to odd gate-induced charge in the QD and integer gate-induced charge in the SI, as away from this gate tuning the excitations turn electron-hole asymmetric, losing their YSR character.

We disagree with the statement of Reviewer #2 that “The YSR and Andreev states appear not only in such superconducting semiconductor junctions, but also in other research areas

like the **surface physics**.” The reason why superconducting Coulomb excitations have not been investigated in surface physics is not because these types of excitations would not also arise in such systems, but most probably because the connection between finite E_c in the superconductor and interesting subgap excitations was not made before.

To make the connection clear, we now explicitly state from the first paragraph of the paper that “Such exploration is of relevance in the study of magnetic impurities adsorbed to superconducting droplets [5,6]”. Fig. 1a of the paper is now an idealized representation of the basic system needed to observe the SCS and Coulomb-aided YSR excitations, which does not necessarily need to be realized on a semiconductor/superconductor junction.

Important questions which can be answered with the STM technique (now included at the end of the main text) are:

1. What is the spatial extension of Coulomb-aided YSR excitations and of SCS excitations? See e.g. Ménard, G. et al. (2015). Coherent long-range magnetic bound states in a superconductor - Nat. Phys., 11(12), 1013–1016.
2. Is there orbital structure in the wavefunctions of these excitations? See e.g. Ruby, M. et al. (2016). Orbital Picture of Yu-Shiba-Rusinov Multiplets - Phys. Rev. Lett., 117(18), 186801.
3. How do YSR states behave in chains of magnetic adatoms when these chains are deposited on top of a $E_c > 0$ superconducting island? See e.g. Schneider, L. et al. J. (2021). Topological Shiba bands in artificial spin chains on superconductors - Nat. Phys., 17(8), 943–948.
4. Do chains of magnetic adatoms on finite E_c superconducting droplets support Majorana zero modes? See e.g. Nadj-Perge, S. et al (2014). Observation of Majorana fermions in ferromagnetic atomic chains on a superconductor - Science, 346 (6209).

I have the following comments:

1. As for Figure 1, I think that the figures a~d are not necessary or not suitable for the explanation of the superconducting Coulombic states. In this paper, the authors mostly compare with YSR, and I think Extended Data Figure 1 is more helpful for understanding than Figure 1.

We have adapted former Extended Data Fig. 1 as part of Fig. 1 as suggested.

2. In Fig. 3b, the change in charge state with respect to V_N is assumed to be one by one. Can this be confirmed from the experimental data? (Is this validated by FigS2 because there is no S_0 expansion in the V_N dependence?)

Reviewer #2 seems to suggest that the QD is proximitized by the superconductor and therefore also superconducting, and thus that it should be charged in steps of 2 electrons instead of 1. The experimental data however confirms that the charge state is changed in steps of 1 by V_N , and not in steps of 2, based on:

- 1) The spin filling pattern (in steps of 1 electron spin) of small-large YSR loops in the regime when $E_{cL}=E_{cR}=0$, shown in Review Fig. 1. The alternation of spacing between conductance lines in the corresponding stability diagram reflects this filling. This pattern is maintained over large changes of V_{SR} and V_{SL} which modify E_{cL} and E_{cR} significantly, indicating that V_N charges the QD by steps of 1 e independently of E_{cL} and E_{cR} (see Review Fig. 2).
- 2) The irregularity of the QD transition lines when V_N is changed, e.g. in Review Fig. 1a. This irregularity occurs due to adjacent QD levels having different tunnel

couplings and different level spacings. If the QD were a proximitized superconductor, this irregularity would not be observed: the SI transition lines are in contrast very regular upon changes of the island gate voltage.

- 3) When Γ is significant so that the 1,1 sectors are developed, these sectors have a markedly different B dependence than the adjacent 0,1 and 2,1 sectors. The 1,1 sectors shrink vertically with B field as shown in Review Fig. 3 for three adjacent QD shells, which would not be possible if the QD were not occupied by one electron to form a YSR singlet GS. When B is increased beyond a threshold, a transition into a triplet GS occurs and the 1,1 sectors expand vertically.
- 4) QD shells can cross each other in B field, as shown in Review Fig. 3 for shells B and C. If the QD were superconducting, we would expect a negligible level spacing and level-crossing would be unresolvable.

Review Fig. 1. Usual YSR regime reached in the supporting SI-QD-SI device by setting $E_{cR}=E_{cL}=0$. The V_N region between -100 mV and -111 mV in these colormaps was tuned in detail in Fig. 5 of the main text (same gate settings). **a** Zero-bias stability diagram. **b** Bias spectrum. The presence of YSR loops alternating in size indicates charge filling of the QD in steps of $1e$. The gate was swept along the dashed line in **(a)**.

Review Fig. 2. 1e charging versus V_N of the QD across large ranges of **a** V_{SR} and **b** V_{SL} , corresponding to large variations of E_{cL} and E_{cR} (with $E_{cL} \rightarrow 0$ and $E_{cR} \rightarrow 0$ towards the top of each plot). ≈ 960 electrons are filled in the right SI in **(a)**, while ≈ 590 electrons are filled in the left SI in **(b)**. The midpoints of the V_{SR} ranges shown in detail in Fig. 5 are indicated by arrows in **(a)**. The V_{SL} setting is in turn indicated by an arrow in **(b)**. At this resolution, Coulomb peaks due to filling of the two SIs are three-pixels wide.

Review Fig. 3. Data acquired in the SI-QD-SI device confirming charging of the QD in steps of $1e$. **a-e** Zero-bias G versus V_{SR} and V_N for $n_L = \text{even}$ at different B field applied parallel to the nanowire. **f** Zero-bias G versus V_{SL} and V_N for $n_R = \text{even}$ at $B=0$, shown for completeness. The appearance of 3 pairs of approximately vertical lines in both **(a)** and **(f)** centered at the same V_N voltages (0.123 V, 0.1285 V and 0.133 V) indicates the presence of a single QD between the two SIs. Charge states n_L, n_N, n_R are indicated in **(a)** and **(f)**. QD shells are identified as A, B and C.

3. In Figure 4, there is a description of the triplet superconducting Coulombic state, but I think it is unclear whether the superconducting island remains "superconductive" when $B=0.35$ T or higher. Therefore, it may be difficult to discuss the phase in $B > 0.35$ T as a triplet superconducting Coulombic state. The data obtained at $B=0.35$ T in Extended Data Figure 7 seems to be very similar to that of Coulomb diamond in the normal state. In addition, (as the authors mention in the manuscript) Se expansion above 0.35 T cannot be found in Fig. 4. If the authors cannot show experimentally that the superconductivity is still there, I think that this argument is too strong.

Experimentally, the presence of superconductivity in our QD-SI and SI-QD-SI devices is established from the difference in sizes of adjacent charge domains with odd and even occupation of the SI, as now indicated in Methods. From these differences, we conclude that the devices remain superconducting up to at least 1.2 T and 1.5 T, respectively, with an upper limit of 2.1 T extracted from S-DQD-S devices.

4. The device configuration is very similar to the previous Majorana signature report (10.1126/science.aaf3961, <https://doi.org/10.1038/nature17162>). In the discussion part, the authors may compare the experimental results with/without the QD or the charging energy in the superconducting island. Why do the authors not indicate such spectroscopic results in the high magnetic field regime? If there are some reasons why the authors cannot obtain the Majorana signature in this device, it is helpful to write them on the manuscript.

We obtain the “Majorana signature” ($1e$ periodicity of Coulomb peaks) in our main device in all Γ regimes investigated; however, our devices were not designed for the purpose of observing Majorana excitations (the SIs are too short and there is an intentional QD next to them, which already produces “trivial” subgap excitations), and therefore we are most certain that the $1e$ periodicity which we observe at large B does not correspond to Majorana excitations in the SI.

We have largely removed the B dependence data in the paper to avoid diluting its main message (stated in the abstract), and to shorten the length of the main text, the number of Extended Data Figures and the length of the Supplementary Information. Therefore, in its present shape the paper does not contain a discussion about the Majorana signature.

We do not have large B data in the YSR regime without the charging energy in the superconducting island shown in Fig. 5a.

We did not attempt to remove the QD through gate tuning.

Reviewer #3 (Remarks to the Author):

The joint experimental and theoretical work focuses on the subgap spectrum of a quantum dot attached to a superconductor island with finite size and where the superconducting gap and the charging energy of the island are comparable. Extensive transport data is presented and there is an impressive agreement between experimental findings and theory. The topic is timely, since superconducting subgap states are at the center of attention in the research of hybrid devices due to their potential in novel qubit. The work is original, based on my knowledge, subgap states have not been studied in such geometry.

The presentation is clear, however it is over concise, the reader should rely on Extended Data Figures to follow the manuscript. Especially content from the first three Extended Data Figures are missing.

We thank Reviewer #3 for her/his useful comments. We now only focus on the effect of the Coulomb repulsion in the superconductor on 1) transforming the Yu-Shiba-Rusinov many-body singlet into a two-body state and 2) rendering the subgap excitations electron-hole asymmetric.

Moreover, we have incorporated content from the first three Extended Data Figures into the main text as text (right after the introduction) or figures (new Figs. 1 and 5).

Altogether I find the work valuable to be published in Nature Communications if the following points are properly addressed:

- Superconducting islands with finite charging energy was addressed in the past e.g. in metallic grains e.g. by D.C. Ralph et al. Short review of these works is missing in the introduction either on the experimental or on the theory side.

The early work on superconducting islands went in two directions:

- 1) The observation of discrete levels in the smallest of the grains and how a large level spacing destroys superconductivity (e.g. D.C. Ralph et al.).
- 2) Cooper-pair filling and even-odd charging effects, which also occur in significantly larger islands (e.g. Tinkham et al.).

Only the second type of effects are relevant for our manuscript, and therefore we have included two references to this end:

Ref. 2, which describes the Cooper-pair filling spectroscopy.

Ref. 3, which describes even-odd effects in hybrid semiconductor-superconductor nanowires.

For conciseness and clarity, we do not mention the first type of effects, as our SIs are too large to show size effects.

- Based on Fig. 1c and on standard double dot physics, it is a rather straight forward expectation to receive an asymmetric spectrum of SCS for $E_C > 0$.

This is correct. However, due to the Yu-Shiba-Rusinov binding, the Cooper pairing, and the even-odd effects in the SI, the exact bias positions of the asymmetric features are beyond standard double dot physics, as we show experimentally in new Fig. 5.

Reading through the conclusion of the paper, the reader is left with a sense of lack, what motivated the study of this particular geometry? It would be desirable to extend the end of the manuscript with a section how this setup could be used for e.g. qubits.

The motivation behind the QD-SI geometry was to find out what happens to the YSR excitations in the presence of E_c in the superconductor. A second motivation was to learn to distinguish between QD-induced subgap excitations and Majorana excitations in a SI.

In turn, the motivation behind the SI-QD-SI geometry was to realize the two-channel Kondo model in the superconducting state. Our experimental results on this new and interesting system are gathered in a forthcoming manuscript.

A more thorough account of our initial motivation for this geometry can be found under "Project Objectives" in the 2017 description of the Marie Curie Skłodowska Individual Fellowship by the first author which partially funded this work: <https://cordis.europa.eu/project/id/832645>.

We have uploaded on arXiv a manuscript (Pavešić, L., & Žitko, R. (2021). Qubit based on spin-singlet Yu-Shiba-Rusinov states. arXiv, 2110.13881) explaining how the SI-QD-SI device used here as a supporting device can be used to obtain two singlets which are the basis for a new YSR qubit immune to magnetic field. An additional manuscript detailing the experimental observation of two singlets with identical total charge (and thus coherent and useful for a qubit) in our SI-QD-SI device is under preparation.

We briefly summarize these motivation points in the introduction and conclusion.

- Spin-orbit interaction generates a non-trivial spin character of electron states. When electrons tunnel from the supra island to the dot this generates a non-spin conserving tunneling between the two natural spin basis. How that would effect the spectrum, is there some experimental signature of this effect?

The SO interaction will mix singlet and triplet states, potentially smoothing crossings of these states, which should be observable in the S_o , S_e versus B dependence. We are currently exploring how the model can account for this in relation to the experimental observations. However, the present version of the manuscript does not include large B data at which these crossings would occur to avoid diluting the main message stated on the abstract.

Does it influence the presented g-factor study?

Yes, the SO interaction should influence the g-factor presented in this study. It is well known that the SO interaction in InAs leads to its large bulk g-factor.

- For completeness please add other parameters e.g. V, Gamma in Fig. 2b.

This figure is about the electrostatics of the device. V is related to C_m as explained in the text, but Γ is not about electrostatics, so it does not belong in this figure. We believe that adding V in this figure will confuse the reader. For clarity and correctness, however, we have added the tunnelling resistors in Figs. 2 and Ext. Data Fig. 5 as it is standard practice for double and triple QD devices.

- Does the reader understand well that the Al superconductor is epitaxial and grown in MBE after wire growth?

Yes, this is indeed the case. We now specify this on Methods.

- It is stated that at B_D in Fig. 4 3 states get degenerate. Please explain clearer in the text how more than 3 states could be degenerate at a particular point. In general it is unlikely.

$B=B_D$ corresponds to B where the triplet state with parallel up spins is pushed down by the Zeeman energy to become degenerate with the singlet state precisely at the gate voltage of the doublet-singlet GS crossing (existing for $B < B_D$). However, in the interest of focus, we have removed this part of the story from the article.

- For Extended Data Figure 3b the caption seems not precise.

Precisions made for the caption of this figure (now identified as Extended Data Fig. 1) are shown in red.

- In Fig. 3b (experiment) FP point has larger conductance as TP. Does it consistent with theoretical expectation?

The theory in this work does not include transport calculations and its output is not differential conductance, so we do not have a concrete expectation. However, given that more states are degenerate in the FP, it would naively be expected that the FP has larger conductance than the TP. In the experiment, however, the FP does not systematically show a larger conductance than the TP. For example, in Fig. 3a the left FP has a similar conductance as the left TP, while the right FP has a larger conductance than the right TP. In

the interest of message focus, the discussion of multi-degeneracy points has been shortened and shifted to the Supplementary Information, p. 2.

REVIEWERS' COMMENTS

Reviewer #1 (Remarks to the Author):

The authors have largely revised the manuscript. They properly addressed all concerns in my previous report and removed ambiguous sentences or figures. The focus of the paper is on subgap excitations in hybrid superconducting-semiconducting nanostructures where Coulomb charging effects can be tuned electrostatically.

The findings and their theoretical interpretation are sound.

The novel figures, in particular Fig. 1, are helpful in understanding the complex manybody physics probed by the experiment.

I can now recommend publication in Nature Communications.

Reviewer #2 (Remarks to the Author):

The authors answered my critics sufficiently. In addition, they clearly explained my concerns about the relation between the charging energy and the Andreev bound states. I appreciate that. However, still I have some questions about their response.

1. In their response against Q.3, they said "Experimentally, the presence of superconductivity in our QD-SI and SI-QD-SI devices is established from the difference in sizes of adjacent charge domains with odd and even occupation of the SI, as now indicated in Methods." and I found in the methods section "In the QD SI and SI-QD-SI devices, the presence of superconductivity at large B was determined from size differences of adjacent charge domains with odd and even occupation of the SI, observed up to $B = 1.2$ T and $B = 1.5$ T, respectively.". However, I cannot find the corresponding figures in the supplementary unless I overlooked them.

(Maybe this is not important in the present version because they removed their discussion about $B > 0.35$ T)

2. About the importance of this manuscript, they improved their introduction section. However, still I am not convinced about the importance. As they mentioned, their results lack electron-hole symmetry. This means that their discovery of the superconducting Coulombic states cannot be applied to realize the topological superconductivity at least. In addition, they mentioned the multichannel Kondo model. I agree that the multiterminal Kondo model is one of the important topics in this research area but I am not convinced why the superconducting variants of that model are important to develop the research area. I think they need to explain the importance of their results and future perspectives more clearly and carefully.

Therefore, I cannot believe that the manuscript is valuable for the publication still.

Reviewer #3 (Remarks to the Author):

Thanks for the revised manuscript, it is significantly improved compare to the original version. I support to publish this work in Nature Communication.

Reviewer #1 (Remarks to the Author):

The authors have largely revised the manuscript. They properly addressed all concerns in my previous report and removed ambiguous sentences or figures. The focus of the paper is on subgap excitations in hybrid superconducting-semiconducting nanostructures where Coulomb charging effects can be tuned electrostatically.

The findings and their theoretical interpretation are sound.

The novel figures, in particular Fig. 1, are helpful in understanding the complex manybody physics probed by the experiment.

I can now recommend publication in Nature Communications.

We thank the reviewer for recommending our work for publication.

Reviewer #2 (Remarks to the Author):

The authors answered my critics sufficiently. In addition, they clearly explained my concerns about the relation between the charging energy and the Andreev bound states. I appreciate that. However, still I have some questions about their response.

We thank the reviewer for the additional comments.

1. In their response against Q.3, they said “Experimentally, the presence of superconductivity in our QD-SI and SI-QD-SI devices is established from the difference in sizes of adjacent charge domains with odd and even occupation of the SI, as now indicated in Methods.” and I found in the methods section “In the QD SI and SI-QD-SI devices, the presence of superconductivity at large B was determined from size differences of adjacent charge domains with odd and even occupation of the SI, observed up to $B = 1.2$ T and $B = 1.5$ T, respectively.”. However, I cannot find the corresponding figures in the supplementary unless I overlooked them.

(Maybe this is not important in the present version because they removed their discussion about $B > 0.35$ T).

Indeed, these measurements are not as important in the present version of the manuscript. For completeness, Review Figs. 1 and 2 below show the B dependence of the odd and even domain sizes up to $B = 1.2$ T in the QD-SI device, and up to $B = 1.5$ T in the SI-QD-SI device. Size differences can be observed up to these fields in each device.

Review Figure 1. **B dependence in the QD-SI device.** **a** Zero-bias conductance versus magnetic field B and superconducting island gate voltage. **b** B dependence of S_o , S_e spacings, extracted from the bars in (a). Error bars correspond to conductance peak widths.

Review Figure 2. **B dependence in the SI-QD-SI device.** **a-e** Zero-bias conductance versus superconducting island gate voltages at different B . **f** B dependence of S_{oL} , S_{eL} , S_{oR} , S_{eR} spacings in the left and right islands, extracted from the bars in (a-e). Error bars correspond to conductance peak widths.

2. About the importance of this manuscript, they improved their introduction section. However, still I am not convinced about the importance. As they mentioned, their results lack electron-hole symmetry. This means that their discovery of the superconducting Coulombic states cannot be applied to realize the topological superconductivity at least.

Topological order is a property of the states, not of their excitations. Therefore, it is irrelevant for the realization of topological order and of topological superconductivity whether the excitations are electron-hole asymmetric or not.

In addition, they mentioned the multichannel Kondo model. I agree that the multiterminal Kondo model is one of the important topics in this research area but I am not convinced why the superconducting variants of that model are important to develop the research area. I think they need to explain the importance of their results and future perspectives more clearly and carefully.

The relation between our devices and the two-channel Kondo model is in the simulation of its intermediate coupling fixed point using recursive QD-SI arrays of odd length. In the discussion section of the manuscript (changes noted in red), we expand on how our devices can be used in the simulation of general spin effects, with schemes of the concepts in new Supplementary Figure 12.

Reviewer #3 (Remarks to the Author):

Thanks for the revised manuscript, it is significantly improved compare to the original version.

I support to publish this work in Nature Communication.

We thank the reviewer for recommending our work for publication.